# The Immune-Centric Revolution in the Diabetic Foot: Monocytes and Lymphocytes Role in Wound Healing and Tissue Regeneration—A Narrative Review

**DOI:** 10.3390/jcm11030889

**Published:** 2022-02-08

**Authors:** Laura Rehak, Laura Giurato, Marco Meloni, Andrea Panunzi, Giada Maria Manti, Luigi Uccioli

**Affiliations:** 1Athena Biomedical Innovations, Viale Europa 139, 50126 Florence, Italy; laurarehak@gmail.com (L.R.); giadamanti@hotmail.com (G.M.M.); 2Diabetic Foot Unit, Department Internal Medicine, Policlinico Tor Vergata Teaching Hospital, 00133 Rome, Italy; lauragiurato@yahoo.it (L.G.); meloni.marco@libero.it (M.M.); andreapanunzi@gmail.com (A.P.)

**Keywords:** wound healing, diabetic foot, immune system, monocytes, lymphocytes, macrophage polarization, tissue regeneration

## Abstract

Monocytes and lymphocytes play a key role in physiologic wound healing and might be involved in the impaired mechanisms observed in diabetes. Skin wound macrophages are represented by tissue resident macrophages and infiltrating peripheral blood recruited monocytes which play a leading role during the inflammatory phase of wound repair. The impaired transition of diabetic wound macrophages from pro-inflammatory M1 phenotypes to anti-inflammatory pro-regenerative M2 phenotypes might represent a key issue for impaired diabetic wound healing. This review will focus on the role of immune system cells in normal skin and diabetic wound repair. Furthermore, it will give an insight into therapy able to immuno-modulate wound healing processes toward to a regenerative anti-inflammatory fashion. Different approaches, such as cell therapy, exosome, and dermal substitute able to promote the M1 to M2 switch and able to positively influence healing processes in chronic wounds will be discussed.

## 1. Introduction

Diabetes is a predominant disease worldwide, with patients developing a wide variety of chronic complications, including Diabetic Foot (DF), characterized also by non-healing ulcers [1]. Current therapies for chronic non-healing diabetic wounds are still far from the optimal solution, with poor healing outcomes in many patients [1]. Nonhealing diabetic wounds produce a huge socioeconomic burden, with an estimated cost of USD 40.5 billion annually, and each amputation procedure can cost well over USD 35,000 [2]. Due to the increasing prevalence of diabetes, the total cost of diabetic ulcer care has also drastically increased in the past 20 years [3].

Extensive research tries to better highlight the diabetic wounds pathophysiology and, in particular, the role of inflammatory cell populations within the wound and how they are modified in diabetes [4].

Different cell populations such as mast cells, neutrophils, lymphocytes, monocytes, macrophages, keratinocytes, fibroblasts, and endothelial cells contribute to different stages of skin wound healing [5]. The myeloid lineage is the main supplier of inflammatory cells populations within the wound environment and plays a crucial role in the reparative phases of wounds [5,6]. It is well known that the natural wound-healing process is a four-stage progression that involves distinct and overlapping phases such as hemostasis, inflammation, proliferation, and remodeling, with different cell populations involved [7]. In contrast to acute wounds, which proceed in a well-timed fashion, chronic wounds fail to heal because they are blocked in the early inflammatory state [8].

In physiological wound healing, after the hemostasis due to platelets aggregation, injured tissues release pro-inflammatory mediators, which are essential for controlling infection, clearing necrotic debris, and the induction of the wound healing process [9]. Several cell types produce a transient connective tissue matrix, new blood vessels, and epithelial closure [7]. Newly formed tissues are remodeled.

Both innate and adaptive immune systems play an essential role in orchestrating all the phases of tissue repair and healing, as shown in both preclinical and clinical studies. The innate system, which consists of monocytes/macrophages, innate lymphocytes, basophils, natural killer (NK), granulocytes, tissue-resident mast cells, and dendritic cells, mobilizes rapidly but with low specificity. On the contrary, the adaptive system, which includes T and B lymphocytes, is activated more slowly with long-term memory and high specificity. The stimulation of innate and adaptive immune responses is activated by damage signals released from apoptotic and necrotic cells, which induce an alteration in the wound microenvironment [10,11]

Monocytes/macrophages play critical roles in host defense, tissue debridement, and cell regulatory functions [3]. Studies in monocyte/macrophage-depleted mice show that these cells are essential for normal wound healing, collagen deposition, angiogenesis, and wound closure [12,13]. The dysregulation of both monocytes/macrophages and unbalanced macrophage phenotypes may lead to impaired or reduced healing [14,15,16,17]. Impaired wound healing in diabetes has been associated with an increased number of wound monocytes/macrophages, as well as an impaired transition from pro-inflammatory into pro-healing wound [18,19]. In addition, a reduced phagocytic ability has been correlated to chronic inflammation in diabetes wounds [20,21]. Monocytes/macrophages are essential, but they do not play alone. Lymphocytes T, and in particular the subpopulation Regulatory T-cells (Treg), have been shown to promote repair and regeneration of various tissue such as skeletal and heart muscle, skin, lung, bone, and the central nervous system [22]. Recent data also suggest an unexpected key role of Treg in the angiogenesis and tissue regeneration in diabetic wound [23].

Recently, it has been recognized the influence of immune system on the regenerative therapies, according to a so-called “immune-centric revolution” or “macrophage centered approach [24,25,26]. For this reason, this review will give a brief insight into innovative autologous cell therapies and biomaterials able to immuno-modulate wound healing processes in a regenerative anti-inflammatory fashion. Examples of several different approaches that have been taken toward promoting anti-inflammatory (M2-like) macrophages to heal chronic wounds will be discussed.

## 2. Macrophage’s Classification: An Overly Complex Issue

During wound healing, process macrophages assume distinct roles to guarantee proper healing [5,6]. Macrophages’ phenotypes evolve along with the different stages of wound healing and can be classified roughly into the M1 class, which represents the classically activated phenotype in a pro-inflammatory state, and the alternatively activated M2 macrophages, which inhibit inflammation [27,28]. The classification of macrophages into M1 and M2 subtypes is a rather basic generalization of a more complex continuum of macrophage subtypes. Some authors describe this scenario as “the macrophage spectrum” in which cells possess varying degrees of M1- or M2-like characteristics [28]. Moreover, macrophages can go back and forth between different phenotypes depending on the cellular environment. In the attempt to classify an overly complex the dynamic macrophages populations, different classifications and nomenclatures exist based on activation, release, and surface markers. In addition, macrophage nomenclature is unclear whether the in vitro observed phenotypes are distinct or even applicable to in vivo wound healing [29]. Moreover, wound macrophages can develop a different phenotype depending on numerous factors, such as the anatomical setting of the wound, the precise area within the wound (center/edge), the environment (moist, dry), and if the wound is infected or not [30].

### 2.1. Macrophages’ Classification Based on Activation Cues and on Cell Surface Markers

Macrophages’ phenotypes change due to spatial-temporal cues during wound healing. Several different subsets of macrophages, beyond the limited over-simplification of M1 and M2, have been defined on their activation, cytokine/growth factor/chemokine release, and cell surface markers [31,32,33]. From the activation point of view, M1 and M2 macrophages can be activated by interferon-gamma (IFN-γ), tumor necrosis factor alfa (TNF-α), and the bacterial wall component lipopolysaccharide (LPS) in the inflammatory type M1. On the contrary, interleukins IL-4 and IL-13 induce the anti-inflammatory type M2. Depending on their activation in vitro, M2 macrophages have been further classified into different subpopulations: The M2a is activated by IL-4 or IL-13; M2b is activated by immune complexes, IL-1β, or LPS; M2c is activated by IL-10 and TGF-β; and M2d predominantly secretes IL-10 and vascular endothelial growth factor or VEGF [30].

Regarding the expression of cell surface markers, M1 macrophages express CD86 while regenerative M2 macrophages express elevated levels of the CD206 marker (mannose receptor). CD206 is a distinguishing surface marker for M2a linked to high release of arginase-1 (in mice), PDGF-BB, IGF-1, and several chemokines such as CCL17, CCL18, and CCL22 [32].

### 2.2. Macrophages Classification Based on Release

Regarding the cytokine/growth factor/chemokine release and in agreement with the multiple phases of wound healing, macrophages in vivo have been classified into three different sub-populations called pro-inflammatory, pro-wound healing, and pro-resolving macrophages.

Pro-inflammatory macrophages present shortly after the wound releases nitric oxide (NOS), Reactive Oxygen Species (ROS), IL-1, IL-6, TNF-α, IFN-γ, and metalloproteinases MMP-2 and MMP- 9 to digest the extracellular matrix [34].

Pro-wound healing macrophages release Platelet-Derived Growth Factor (PDGF), insulin-like growth factor 1 (IGF-1), VEGF, and Transforming Growth Factor-beta (TGF-β1) in high concentration to induce cellular proliferation, granulation tissue formation, and angiogenesis [35]. To counteract MMPs and permit ECM formation, pro-wound healing macrophages release tissue inhibitor of metalloproteinases 1 (TIMP1) [35].

Pro-resolving macrophages, in the last wound healing phase, suppress inflammation, releasing IL-10 together with arginase 1 and TGF-β1. Pro-resolving macrophages also release MMP-12 and MMP-13 to remodel and reinforce the ECM, aiming to restore tissue homeostasis and reduce fibrosis [34,36]. Pro-inflammatory and pro-resolving macrophages display some similar features as their actions overlap in the proliferation and remodeling phases. M2a macrophage sub-populations produce collagen precursors and growth factors to stimulate fibroblasts and secrete elevated levels of PDGF, which is implicated in angiogenesis [33,37,38]. M2b macrophages are characterized by CD86, CD68, and MHCII surface markers [32]. M2b macrophages reduce inflammation by releasing anti-inflammatory cytokines such as IL-10, IL-6, IL- β, and TNF, NOS, as well as several different MMPs. In vitro macrophages adopt the M2b phenotype after the neutrophil’s phagocytosis [32]. M2c macrophages express CD206, MERTK, and CD163; are stimulated by glucocorticoids, IL-10, and TGF-β; and produce elevated levels of IL-10, MMP-9, IL-1β, and TGF-β and low levels of IL-12 [39]. M2c, sometimes described as deactivated macrophages, are analogous to pro-resolving macrophages. They can evolve from M1 macrophages with a “deactivated” gene profile to polarize in M2c macrophages. M2 macrophages can polarize in all a, b, and c phenotypes [32]. M2d macrophages stimulated by IL-6 and adenosine do not express either CD206 (mannose receptor) or dectin-1. They produce a high concentration of vascular endothelial growth factor (VEGF), IL-10, and TGF-β while downregulating TNF-α and IL-12 to dampen inflammation [30].

## 3. The M1/M2 Wound-Healing Paradigm: The Switch from KILL (M1) to HEAL (M2)

In the first phase of healing, just after hemostasis, pro-inflammatory M1 macrophages infiltrate after injury to clean the wound from bacteria, dead cells, and foreign debris [31]. When the tissue begins to repair in acute wounds, the overall macrophage population switches to the M2 phenotype, which induce anti-inflammatory and regenerative effects. It is well documented that such transition of phenotypes, defined by the term “polarization”, is an essential step for wound healing [31,32].

The M2 polarization event induces the migration and proliferation of fibroblasts, keratinocytes and endothelial cells to repair the dermis, epidermis, and vasculature [37,38], and this cross-talk is impaired in diabetic wounds [39]. During this phase, both M1 and M2 are also responsible for the vascularization process, first creating new vessels trough sprouting [40], then creating anastomoses between newly formed vessels [41]. The macrophages’ ability to create a functional anastomosis has been observed in in vivo time-lapse imaging, showing that a macrophage arrives at the lesion, extends filopodia or lamellipodia to physically adhere to vessels’ endothelial ends, and through direct physical adhesion and mechanical traction repairs brain vasculature rupture [42]. The macrophage-mediated repair is conserved also in peripheral blood vessels [42]. This conclusion has been confirmed by the observation that macrophages secrete high concentrations of vascular endothelial growth factor (VEGF)-C to stabilize tip cell fusion and increase vascular complexity [39]. Gurevich et al., in an elegant experiment, showed, through in vivo imaging, that after tissue injury in both mice and zebrafish, macrophages could form angiogenic sprouts and drive neo-angiogenesis and consequent vessels remodeling [43]. This paper also shows that in an in vitro human co-culture model, specifically pro-inflammatory M1 macrophages are essential to initiating sprouting angiogenesis via the targeted delivery of proangiogenic cytokines and VEGF but also that a temporal phenotypic switching to M2 is a requirement to permit appropriate later vessel remodeling and regression [43].

In the final remodeling phase, macrophages release metalloproteinases (MMPs) to digest the temporary extracellular matrix, and then they start going into apoptosis [44]. In chronic wounds, pro-inflammatory macrophages persist in the M1 phenotype without transitioning to M2 anti-inflammatory phenotypes, which is believed to contribute to tissue repair impairment [36,45,46,47]. Therefore, controlling the phenotypic switch from M1 to M2 could represent a favorable solution for the transition from the inflammation to the proliferation stage of wound repair.

It is not clarified if the transition from M1 to M2 phenotype occurs through neo-differentiation of newly recruited monocytes from peripheral blood or/and through direct polarization of existing resident macrophages in situ to an anti-inflammatory phenotype. It has been observed that this switch can be driven by environmental changes in cytokines, miRNAs, transcription factors, exosomes [48,49], and the modulation of pro-inflammatory and anti-inflammatory receptors [50]. In the injured tissue, efficient clearance of apoptotic cells by wound macrophages (efferocytosis) is a requirement for inflammation resolution. Emerging evidence indicates that microRNA-21 (miR-21) may regulate the inflammatory response promoting efferocytosis [51]. It has been recently demonstrated that M2-derived exosomes can induce direct reprogramming of M1 into M2 with almost 100% conversion effectiveness [52]. These new reprogrammed polarized M2 macrophages produce matrix metalloproteinase (MMP) and vascular endothelial growth factor (VEGF), and their role is essential for angiogenesis re-epithelialization in the proliferative phases [52].

## 4. Resident Dermal Macrophages and Circulating Monocytes-Derived Macrophages

Macrophages can also be divided into two main groups according to their origin: (a) a resident tissue macrophage (RTM) population, called dermal macrophages, which are self-renewing cells derived from the embryonic yolk sack and established before birth, and (b) circulating monocytes that are recruited to injured tissue and differentiate into macrophages [34]. Skin wound macrophages originate from tissue-resident macrophages and infiltrating monocytes, with significantly higher contribution from the circulating monocytes group [38]. Resident dermal macrophages respond to wounds through the recognition of molecules called damage-associated molecular pattern (DAMP) or, in case of infection, pathogen-associated molecular pattern (PAMP) releasing ROS, which in turn initiates a pro-inflammatory cascade [38]. Dermal macrophages also recruit neutrophils to fight the infection [38]. The main goal of resident macrophages is to retain skin integrity, homeostasis and tissue repair [53]. Characteristic surface markers of dermal macrophages are CD64+, MERTK+, and CCR2-/low. They show a high phagocytic ability but a slow turnover [54]. After remodelling, dermal macrophages self-renew and clear apoptotic cells in the resolution phase to return to tissue homeostasis [55,56]. Mapping studies have uncovered that most RTMs are principally of prenatal origin (yolk sac or fetal liver), while monocytes are constantly produced by hematopoiesis [57]. Through a combination of dynamic intravital imaging and confocal multiplex microscopy, it has been demonstrated that tissue-resident macrophages through a ‘‘cloaking’’ mechanism prevent neutrophil-mediated inflammatory damage, maintaining tissue homeostasis [57]. Failure of this cloaking process led to unrestricted inflammatory reactions, neutrophil swarms, and collateral tissue damage that required consequent control of neutrophil-driven inflammation by the recruitment of further circulating monocytes. RTMs represent a previously unknown immune checkpoint to prevent constant inflammatory damage.

Circulating monocytes infiltrate tissues upon initiation of the inflammatory cascade, where they can become definitive macrophages [58]. In vivo imaging showed that an initial wave of monocytes enters the wound simultaneously with neutrophils and not in a second time, as previously thought [59]. While resident macrophages initiate the local inflammatory response with short-term effects, monocyte-derived macrophages are recruited from peripheral blood for hours (about 24 h in mice) after the tissue damage [60]. The monocyte-derived macrophages’ recruitment is due to signals from damaged tissue, both via DAMPs or PAMPs [61]. A common PAMPS is a lipopolysaccharide (LPS), a component of Gram-negative bacteria’s outer membrane, which is recognized via binding by toll-like receptor 4, which activates NF-κB, which in turn induces the expression of pro-inflammatory genes [61]. Examples of DAMPs are extracellular DNA, RNA, and ATP released from dead cells, which attract immune cells to the injury sites [62]. Monocytes can also be recruited efficiently by chemokines and cytokines such as IL-1, IL-6, TNF-α, and MCP-1 (CCL2) [63]. In particular, MCP-1 plays a vital role in the inflammatory angiogenic response by recruiting host monocytes from the blood into the ischemic damaged tissue [64,65,66]. Moreover, MCP-1 promotes healing in diabetic wounds by restoring the macrophage response [66].

Resident and monocyte-derived macrophage coexist in the same wounds, and can show different states of activation. Two distinct macrophage subsets in skin wounds with distinct functions and origin have recently been demonstrated: CX3CR1hi macrophages are derived from tissue-resident macrophages and were predominantly activate in M2, while CX3CR1–/lo wound macrophages are derived from recruited monocytes and exhibit both activation phenotypes M1/M2 [67]. Migratory monocytes populate peripheral tissues in meaningful numbers and cooperate actively in tissue-protection with RTM. RTM play roles in primary prevention of inflammation and recruited monocytes in secondary resolution [57].

### Circulating Monocytes and Their Role in Wound Healing

Peripheral blood monocytes are present in two categories: pro-inflammatory (“classical” monocytes, surface marker CD14+CD16− in human and Ly6C+/high in mice) and anti-inflammatory (“non-classical” monocytes, surface marker CD14low/−CD16+ in human and Ly6C−/low in mice), which are attracted to the injured tissue [61].

Olingly et al., by means of selective labelling, demonstrate that circulating non-classical monocytes are directly recruited within wounds, where they home to a perivascular niche and generate M2 wound healing macrophages [61]. Moreover it has been observed that the local delivery of a small molecule (FTY720) able to recruit non-classical monocytes supports vascular remodeling after injury, confirming an angiogenic roles of peripheral blood monocytes [61]. Blood-derived, non-classical monocytes are major contributors to alternatively activated M2 macrophages, highlighting them as key regulators of inflammatory response and regenerative outcome.

The pro-inflammatory monocytes (CD14+CD16− in human, Ly6C+/high in mice) are derived from spleen and bone marrow, and their concentration increase in the peripheral blood after an injury; when there is no injury, they do not tightly adhere [32].The numbers of pro-inflammatory monocytes showed a short half-life (only 20 h in mice), vary depending on new cells recruitement from the bone marrow and from peripheral blood circulation, and reach a peak ~48 h after injury, while the recruited anti-inflammatory monocytes have a longer half-life (>2 days, in mice) [32]. Anti-inflammatory monocytes attach to the blood vessel wall via α_L_β_2_ integrin (LFA-1) and L-selection (CD62L), which enables anti-inflammatory monocytes to crawl on the endothelium, even during homeostasis, so that they are ready to repair tissue and promote vascular repair when needed [32,68]. These data suggest that, in adjunct to resident tissue macrophage RTM, “resident” monocytes may also be present.

Other authors have used a different nomenclature to group human monocytes into three group: classical (CD14++CD16−), intermediate (CD14dimCD16++), and non-classical (CD14++CD16+) phenotypes. The “classically activated” CD14++CD16− monocyte phenotype represents 85% of circulating monocytes in normal healthy individuals. In comparison, the remaining 5% of the monocyte population is represented by the intermediate CD14dimCD16++ and 10% by the “non-classically activated” CD14++, CD16+ phenotype [60]. In inflammatory environments, classical monocytes differentiate into M1 macrophages, while non-classical monocytes differentiate into M2 macrophages to help in tissue repair [5,6]. Classical inflammatory monocytes are recruited to wounds in a higher amount following injury compared to non-classical monocytes, but there are evidences that inflammatory monocytes can become anti-inflammatory monocytes and differentiate into M2 macrophages [32,69]. This was also confirmed from Arnold et al. in an animal model of injured skeletal muscle, where recruited monocytes exhibiting an inflammatory profile that operates phagocytosis rapidly are able to convert into anti-inflammatory M2 macrophages, which in turn stimulates myogenesis and fiber growth [70]. Interestingly, it was recently observed that the implant of autologous peripheral blood mononuclear cells, produced by a point of care device based on selective filtration for human use, in a non-healing diabetic wound induced the polarization from M1 to M2 and the complete healing [71]. Inflammatory monocytes of the CD14++, CD16+ phenotype are strongly increased in ageing and chronic inflammatory disease [72]. Recently, a study has demonstrated that circulating CD16++ monocytes are a potential biomarker to predict the outcome of diabetic foot wound healing: In peripheral blood, the percentage of CD16++ monocytes and MMP-3 were higher in healed vs. unhealed patients [72].

## 5. The Secret Life of Lymphocytes

While the innate immune system is recognized to play a key role in the tissue healing process, the adaptive immune system has only recently emerged as a key player. Lymphocytes T and in particular regulatory T-cells (Treg) have been shown to promote the repair and regeneration of various tissues [22,73,74,75,76]. Treg can indirectly regulate regeneration by promoting their apoptosis neutrophils, regulating helper T-cells, and inducing macrophage polarization [77,78]. Additionally, Treg can also directly facilitate regeneration triggering resident stem cells locally [79]. Treg can improve the differentiation of stem or progenitor cells such as satellite cells to replace the damaged skeletal muscle and enhance the proliferation of neonatal cardiomyocytes for functional regeneration [80].

Tregs could regulate macrophage polarization through the suppression of IFN-γ produced by CD4^+^ effector T cells, IFN-γ being a promoter of the formation of M1 macrophages. Another mechanism could be related to the increase in IL-10 levels in muscle [81].

In many tissues, Treg are recruited to the injured site to expedite inflammation resolution and to regulate immunity after damage [80]. Tregs promote repair in various tissue: muscle repair after cardiac injury, skin epithelial stem cell differentiation and wound healing, enhance satellite cell expansion in muscle, facilitate lung resolution, promote myelin regeneration in central nerve system, and protect kidney injury [82].

In vitro monocytes in co-culture with Treg produce diminished levels of TNF-α and IL-6 in response to LPS, while Treg alone secretes higher concentrations of IL-10, IL-4, and IL-13 [83]. Treg can also directly act on the pro-inflammatory M1 macrophages through the release of IL-10, inducing the polarization to anti-inflammatory and pro-repair M2 [84]. It has also been reported that diminished levels of Treg facilitate vascular inflammation [81]. Tregs play a highly broad variety of tissue regeneration roles such as facilitating blood flow recovery after ischemia, controlling adipose tissue inflammation, promoting muscle repair, and maintaining tissue/organ homeostasis [77]. Treg can also facilitate cutaneous wound healing, mainly by the secretion of anti-inflammatory/immunosuppressive cytokines, including IL-10, IL-35, and TGF-β [78]. Highly activated Tregs accumulate in skin early after wounding, decreasing both IFN production and proinflammatory M1 macrophage accumulation through the induced expression of the epidermal growth factor receptor (EGFR) [78]. In addition to their regeneration ability, Tregs also promote angiogenesis in ischemic tissue though apelin-mediated sprouting in diabetic patients [79,82]. In keeping, the lack of lymphocytes impairs macrophage polarization and angiogenesis in diabetic wound healing [23]. Tregs are not the only the lymphocyte population able to play a key role in wound healing and angiogenesis: It has been observed that both NK lymphocytes land CD4-T-cells modulate arteriogenesis in a murine ischemia model [85]. Accordingly, an impaired arteriogenic response has been observed in hindlimb ischemia in CD4-Knockout mice [86]. CD8+ T-cell plasticity seems to regulate vascular regeneration [87]. B cells can differentiate into antibody-producing plasma cells but can also present antigens to T cells and modulate local immune responses through the secretion of pro- and anti-inflammatory cytokines. Recent data have shown that naïve B lymphocytes injected in acute or chronic diabetic skin lesions can act as effective modulators of tissue regeneration, both in acute and chronic diabetic skin lesions, accelerating wound healing [88]. The same paper also showed that B cell treatment was associated with better-quality collagen deposition and reduced scar formation. The enhanced healing was reinforced by a higher fibroblast’s proliferation, together with a diminished level of apoptosis, a regenerative modulation of cytokines, and matrix metalloproteinases. The same process was not observed by the injection of disrupted B cells or hematopoietic stem cells. All these relevant insights suggest a potential development of innovative cell therapies based on immune system cells such as monocyte/macrophages and lymphocytes to target vascular diseases associated with diabetes.

## 6. Monocytes/Macrophages and Lymphocytes in Diabetic Wound Healing

In a healing wound (Figure 1), just after neutrophil recruitment, a first wave of monocytes invades the tissue and differentiates into inflammatory M1 macrophages, releasing cytotoxic and proinflammatory molecule such as IL-1β, TNF-α, IL-6, and ROS, with the aim to digest damaged cell, microbes, and necrotic damaged tissue. The first wave is followed by a second wave of anti-inflammatory M2 macrophages, which, on the contrary, promote tissue remodeling, fibrosis, and wound healing through the release of TGF-β, IL-10, and other anti-inflammatory cytokines. Primarily, these M2 populations produce growth factors, anti-inflammatory mediators, although keeping the ability to clear apoptotic cells. Moreover M2s recruit endothelial stem cells and promote angiogenesis in the healing wound, allowing the development of granulation tissue and neovascularization [89].

In contrast, diabetic wounds have many structural and functional differences, such as reduced angiogenesis, which produces a hypoxic wound environment and oxidative stress [20]. Diabetic foot ulcer (DFU) healing (Figure 2) does not progress through phases and is characterized by a stalled non-healing state that includes deregulated inflammation that is considered less effective to facilitate progression of healing, reduced angiogenesis, non-migratory epithelium, low response to growth factors, and fibrosis [8]. In the following paragraphs, we will describe the different behaviors of neutrophils, monocytes/macrophages, and lymphocytes in a diabetic wound.

*Neutrophils in diabetic wound:* A high neutrophil count within the wound and the increased neutrophil-to-lymphocyte ratio is well recognized as a characteristic of impaired diabetic wound healing [90]. As part of their antimicrobic defense, neutrophils form extracellular traps (NETs) by releasing decondensed chromatin lined with cytotoxic proteins. Unfortunately, NETs, can even cause tissue damage. Neutrophils isolated from type 1 and type 2 diabetic humans and mice were primed to produce NETs (a process termed, NETosis), and this phenomenon is responsible for delayed wound healing [91]. Accordingly, a previous study showed that a decreased ability of neutrophils to undergo NETosis led to accelerated wound closure [92,93]. This persistent neutrophil activation and induction of NETosis results in the production of further inflammation, while in the normal process, the inflammation resolution is produced by neutrophils’ apoptotic body phagocytosis from infiltrating monocytes/macrophages [94]. In healing wounds, the uptake of apoptotic neutrophils resolves the inflammatory phase by limiting inflammatory cell infiltration and shifting the production of eicosanoids from pro-inflammatory to anti-inflammatory mediators [95]. In diabetic wounds, the inflammatory phase is significantly prolonged by the disruption of mechanisms which both control the influx of neutrophils as well as regulate their inflammatory processes [91]. Accordingly, it has been observed that the transcription factor FOXM1, responsible for activation and recruitment of inflammatory cells, is downregulated in diabetic patients [96].

*Monocytes-macrophages in diabetic wound:* Dysregulations of the inflammatory phase seems to be associated to epigenetic polarization of innate immune cell pro-inflammatory function prior to wound infiltration, probably due to hyperglycemia [20]. The polarization of innate immune cells towards inflammatory phenotypes is correlated to the systemic inflammatory response observed in both diabetic patients and animal models [97]. Recent results in a mice diabetic model suggest that the combination of improved neutrophils numbers with reduced macrophages numbers, monocyte-derived Langerhans cells, and dendritic cells and eosinophils produces an imbalance in the immune cell composition, which may contribute to their impaired healing [98]. While in the healing wound the infiltrating monocytes differentiate into classically activated inflammatory M1 and alternatively activated M2 macrophages, in the diabetic wound, there is a strong polarization into the M1 phenotype, and the switch from M1 to M2 is heavily impaired [18,19,20,36,46,47,48]. Therefore, a lower number of M2 macrophages and a higher M1:M2 ratio will release low levels of growth factors PDGF, FGF, and VEGF and of anti-inflammatory cytokines such as IL-10, TGF-α, and TGF-β, all of which induce the proliferation phase and the effective regulation of inflammation [9]. It has been also observed that M1 macrophages release high concentrations of TNF-α in diabetic rats, and an in vitro high glucose environment facilitates M1 polarization, which are both detrimental to keratinocyte migration [99]. It has been recently shown that negative pressure wound therapy by suppressing autophagy and macrophage inflammation in a mouse model promotes wound healing [100]. Monocytes and macrophages are known to play important roles in neovascularization during wound healing [38,101]. Since macrophages are a central source of VEGF and other angiogenic molecules in wounds, the macrophage deficit may be linked to the documented decrease in wound angiogenesis that is seen in diabetic wounds [102]. A reduced VEGFR1 signaling in the diabetic wound tissue could contribute to impaired angiogenesis [103]. In addition, M2 macrophages boost wound angiogenesis both by direct (macrophage-to-endothelial cell adhesion) and by indirect mechanisms (paracrine effect) [104]. An extremely critical indicator of an effectively healing wound is an efficient controlled proliferative phase produced by an effective angiogenesis together with complete re-epithelization of the wound. The proliferative phase is characterized by granulation tissue formation, which comprises of different cell populations such as fibroblasts, as well as immune cells, together with the formation of new capillaries, which allow epithelial cell migration towards the wound surface in the process of re-epithelization. Unfortunately, in diabetic wounds, monocyte polarization towards M2 macrophages is strongly reduced, while inflammatory phenotype M1 polarization is elevated, and this causes a poor angiogenesis. Moreover, vasoreparative dysfunction has been observed in diabetic CD34+ stem cells due to impaired autocrine/paracrine function and reduced sensitivity to hypoxia, while the injection of freshly isolated circulating CD14+ monocytes into the ischemic limbs of diabetic mice improves healing and vascular growth, suggesting an important angiogenesis potency of monocytes population even in diabetic patients [105,106,107].

*Lymphocytes in diabetic wound:* A recent study underlines the key role of lymphocytes in both diabetic and non-diabetic non-healing wounds [23]. A lack of lymphocytes compromises wound healing in diabetic as well as in non-diabetic mice. Moreover, the pattern of diabetes plus a lack of lymphocytes further worsens the wound, indicating that when the innate regulatory function is missing, unbalanced M1 polarization, inadequate angiogenesis, and reduced wound healing are exacerbated [23]. Recently, it has been observed that the ischemic tissues of type-2 diabetic patients and mice have significantly more CD8+ T-cells than that of their respective normoglycemic counterparts [87]. The systemic inflammation observed in diabetic patients could limit the migration of Tregs and increase the infiltration of Th17 inflammatory population cells able to promote promote neutrophilic infiltration in the diabetic wound, explaining the prolonged inflammatory phase. Notably, the healing process of diabetic wounds may be accelerated by topical retinoic acid, in this manner inducing T cell plasticity and the differentiation of Th17 cells towards Tregs, confirmimg the crucial role of T cells in the regulation of the inflammatory phase of diabetic wound healing [94]. It has also been observed in diabetic patients that the vascular density is negatively associated with CD4+T cells numbers after ischemic injury, while Tregs injection in the ischemic muscle increased vascular density and induced de novo sprouting angiogenesis through a paracrine effect [79]. A decrease in Natural Killer lymphocytes and high IFN-γ levels are correlated to diabetic foot complications and seem to have potential roles in predicting the infection of diabetic foot ulcers [108].

*Keratinocytes and fibroblasts in diabetic wound:* Chemokine Ccl2 secretion by epidermal keratinocytes is directly coordinated by Nrf2, a leading transcriptional regulator of tissue regeneration, that is activated early after cutaneous damage [39]. In diabetic wounds, Nrf2 fails to activate keratinocytes [39]. Keratinocyte-derived Ccl2 promotes macrophage EGF production, which induces keratinocyte proliferation to promote wound repair [39]. A significantly reduced skin resident cell proliferation as well as stem and progenitor cell activation was observed in diabetic foot ulcers, and it seems to be related to multiple factors such as glycation of proteins, reduced angiogenic capability, and oxidative stress, contributing to an extension of the proliferative phase [20]. Fibroblasts in impaired wounds showed ECM deposition significantly reduced abilities. DFU-derived fibroblasts were noted to produce ECMs twofold thinner than normal fibroblast, also showing a superior composition of collagen type I and fibronectin content [109].

## 7. The Immune-Centric Revolution: The Long and Winding Road from Stem Cells to Immune Cells Populations in Regenerative Medicine

Although stem cells have been considered promising for the treatment of degenerative diseases by ‘seeding’ them into damaged tissues, it has recently been observed that the regenerative capacity of stem cells is influenced and regulated by the local immune response and in particular by macrophages, which constitute a central component of the damage response and are the coordinators of tissue repair and regeneration [24]. Among the panoply of immune cells involved in the response to both acute and chronic wounds, recent discoveries have highlighted novel and often unexpected roles for certain types of immune cells in promoting a permissive local environment for effective cell replacement and restoration of tissue integrity. Some studies have shown that the control of inflammation is crucial in regenerative therapies: To be effective, regenerative therapies must block and control inflammation to allow tissue regeneration by resident stem cells [110]. Indeed, the presence of inflammation inhibits the regenerative action of tissue-resident mesenchymal stem cells (MSCs) [110]. Recent papers suggest that an innovative regenerative strategy could be to polarize macrophage from the M1 inflammatory state to the M2 anti-inflammatory state utilizing immune cells [25,32,111,112]. These reviews conclude that next-generation regenerative therapies need an immune-centric approach instead of the use of stem cells. Thus, depending on the tissue or organ targeted, regenerative strategies could be developed to stimulate macrophage polarization or to recruit subpopulations of pro-healing macrophages. Already, Mordechai in 2013 [101] and Pinto in 2014 [102] (7) have shown that the regeneration of myocardial tissue after ischemia was induced by macrophages that regulate resident stem cells and promote regeneration, suggesting that targeting macrophages could be a new strategy to improve infarct healing and repair. The regenerative and stem-cell-controlling capacity of macrophages has also recently been demonstrated in bone tissue by Gibon et al. [103], Gullard et al. [104], and Ekstrom [105]. Najar et al. [106] in 2018 clarified that mesenchymal stem cells act through a paracrine and immune-modulatory and non-differentiative mechanism and that the microenvironment and immune system regulate the activity of MSCs regardless of the tissue from which they originate. Based on the role played by several types of macrophages and lymphocytes in the wound-healing response, it is tempting to hypothesize that interventions that reduce the M1 macrophage phenotype and promote M2 may represent a new therapy to heal chronic wounds.

### 7.1. How to Switch to M2 Regenerative Phenotype?

Macrophages play an important role in wound healing, and the switch to anti-inflammatory M2 phenotypes is necessary for efficient healing. Questions remain regarding monocytes recruitment and macrophage differentiation, specifically whether monocytes are predetermined to differentiate in one specific phenotype, M1 or M2, or if macrophages polarize from M1 to M2 phenotypes (or vice versa) within the wound. Various approaches have been taken to immune-modulate macrophages to polarize in M2 phenotypes and/or simultaneously M1 macrophages (Figure 3). A list of methods to switch to M2, such as immune cell-based therapy, MSC, M2 exosome, and dermal substitute will be discussed.

### 7.2. Immune-Cell-Based Cell Therapy

Cell-based therapies are rapidly emerging in regenerative medicine as dynamic treatments that perform multiple therapeutic functions. Monocytes and macrophages, as innate immune cells involved in inflammation control and tissue repair, are increasing popular clinical candidates due to their angiogenic, anti-inflammatory, and regenerative ability. Table 1 shows a brief description and clinical result of clinical trials based on macrophages or peripheral blood mononuclear cells describe in this review. The treatment of chronic ulcers with blood-derived macrophages activated by hypo-osmotic shock has been used effectively in over 1000 patients in Israel [107]. Previously, Danon et al. in 1997 treated pressure ulcers in elderly patients by injecting macrophages from blood units of young, healthy donors near the wound periphery plus a portion of the cell suspension deposited on top of the wound [113]. Patients were treated with a single implant, or with a second one when delayed healing was present after 1 month later, and wound healing was compared with conventional methods (debridement, antibiotics, and wound dressings). In the macrophage-treated group, 27% healed, while only 6% healed in the control group (*p* < 0.001). Moreover, the macrophage-treated group showed a faster healing (*p* < 0.02), and no side effects were reported [113]. A second prospective controlled trial was designed to compare macrophage injections from healthy donors (66 patients) to standard care treatments (38 patients) for stage III and IV pressure ulcers in elderly patients. The results showed a significant higher percentage of completely closed wounds in the macrophages-treated group in comparison to standard care [114]. Interestingly, in the subset of diabetic patients 65.5% of wounds with the macrophage treatment healed, while only 15.4% of healing was observed in the standard care group [114]. Magenta et al. recently published an extensive review on autologous cell therapy from different tissue sources (blood, bone marrow, and adipose tissue) to treat critical limb ischemia in diabetic patients, reporting data from basic science to clinical trials [115]. Autologous cell therapy, in particular, autologous Peripheral Blood Mononuclear Cells (PBMNC), based on monocytes/macrophages and lymphocytes represent an interesting strategy to treat non-option critical limb patients and diabetic foot patients [116,117,118,119,120,121]. Rigato et al. on a recent meta-analysis on no-option critical limb ischemia (NO CLI) patients showed that PBMNCs, but not other cell types, were associated with a significant decrease in amputation and increase in amputation-free survival [122]. The same results were observed by Liew et al. in a meta-analysis of 16 randomized trials where PBMNC lowered the risk of major amputation and increased ulcer healing significantly [123]. Three other meta-analyses on autologous cellular therapy including PBMNC on diabetic foot patients showed a benefit of wound healing and reduced amputation associated with TcPO_2_ increase and reduced pain [124,125,126]. Dubsky et al. have treated 28 patients with diabetic foot disease (17 treated with bone marrow cells and 11 with PBMNC) comparing the result with a control group treated with standard care at 6 months and have reported a statistical increase in TcPO2 with no significant differences between bone marrow cells and peripheral blood cell groups, while no change in transcutaneous oxygen pressure in the control group was observed [119]. In addition, the 6 month major amputation rate was significantly lower in the cell therapy group compared with that in the control group (11.1% vs. 50%), with no difference between bone marrow cells and peripheral blood cells [119]. Interestingly, the same group reported a comparable improvement of CLI major amputation with autologous cell therapy in diabetic foot patients compared with repeated PTA and a more effective healing of foot ulcers in the cell therapy group [127]. A user-friendly point of care device based on peripheral blood selective filtration to be used for intra-operatory use in human cell therapy has been developed to produce fresh autologous PBMNC, with evidence in terms of adequate potency in therapeutic angiogenesis in vitro and in vivo [128]. Promising results have been obtained from implanting PBMNC produced by a specific device (Hematrate Blood Filtration system Cook Regentec) in different clinical trials including diabetic patients [120,121,129,130]. Persiani et al. have observed a 9.4% decrease in major amputation in 18 no-option patients with diabetes treated by PBMNC together with an increase in TCPo2 and a pain reduction at 2 years [120]. A similar result in terms of major amputation has also been previously reported on CLI non-option patients, including diabetic and Burgers patients, treated with PBMNC produced by apheresis [116]. Interestingly, it has been demonstrated by a histological examination of incisional biopsies of diabetic non-healing ulcers that autologous PBMNC implants produced by this selective filtration point of care and injected perilesionally around diabetic non-healing wounds polarize M1 macrophages in M2. Moreover, the implantation of A-PBMNC promotes relevant changes in the overall molecular setting over time [71]. The consequent cellular and biochemical adaptations favor the establishment of conditions similar to physiological ones that progressively support the regeneration of damaged tissues and finally wound healing measured as inhibition of HIF, NF-KB, and TNF-alpha, progressive polarization of M1 into M2, increase in VEGF, and newly formed capillaries [71]. As the regenerative processes occur, an increase in the vascular network formation is clearly seen [71]. These preliminary data confirm in the ability of fresh, naïve, autologous PBMNC to induce immunomodulation through macrophage polarization and that this results in complete wound healing in a diabetic ulcer. On the contrary, the delivery of macrophages polarized in vitro into M2a and M2c phenotypes and then injected into mouse wounds did not accelerate healing in wild type mice and delayed healing in diabetic mice [131]. The same study also observed a delayed re-epithelialization and persistence of neutrophils and M2 macrophages in diabetic treated wounds 15 days post-injury, suggesting that the application of ex vivo generated M2 macrophages is not beneficial and contraindicated for cell therapy of skin wounds. It seems instead that to produce a positive clinical outcome in terms of wound healing, polarization should occur in the patients in the wounded tissue which send the right microenvironmental signals to PBMNC. The same groups showed that the implants of Matrigel supplemented with M2a and M2c macrophage subsets in a mice wound model showed an increased number of endothelial cells and tubular structures, while M1-enriched Matrigel did not, suggesting that macrophages polarized towards an M2 phenotype seem to have a higher angiogenic potential compared to other subsets [132]. Accordingly, Di Pardo et al. also observed an increase in VEGF and laminin in the diabetic wound after PBMC implant [71]. A similar results was observed for the first time by De Angelis et al. in no-option CLI patients, including a subset of diabetic patients, after PBMNC implant [121]: histological data confirmed dermal granulation tissue and an increased number of monocytes (CD68+) and newly formed micro vessels (CD31+). After the PBMNC treatment in the healed epidermis, the presence of the new vessels was observed, whereas dermal inflammation and monocyte infiltration were reduced. All these data suggest that autologous PBMNC represent a safe and effective therapy for diabetic foot non-healing wounds. Considering the low invasiveness and the repeatability, PBMNC could represent the new frontier that will replace stem cell therapy.

### 7.3. Mesenchymal Stem Cells (MSC)

Accumulating evidence suggests that mesenchymal stem cells (MSC) promote tissue repair through the immune-modulation response and the secretion of growth factors rather than by the substitution of damaged cells [133]. MSCs release a wide range of factors, including PGE2 and interleukin-6 (IL-6), that polarize to a M2 pro-resolving profile [134]. The immuno-regulatory capacity of MSC depends on a process of “licensing” that implies the activation of MSC by the inflammatory environment. The requirement of MSC activation to induce immunoregulation is supported by data showing that the suppression of lymphocytes T proliferation induced by MSC in co-cultures is achieved only after the supplement of adequate amounts of IFN-γ and TNF-α [135]. By producing a large number of immunomodulatory molecules such as TGF-β, hepatic growth factor (HGF), nitric oxide (NO), indolamine 2,3-dioxygenase (IDO), L-10, IL-6, IL-1 receptor antagonist (IL-1Ra), hemeoxygenase-1 (HO-1), prostaglandin E2 (PGE2), and pro-angiogenic factors VEGF, angiopoietin-1, placental growth factor (PGF), HGF, basic fibroblast growth factor (bFGF), TGF-β, PDGF, and IL-6, MSCs regulate immune response and vasculogenesis, crucially contributing to the enhanced repair of injured tissues in various organ [136]. The transplantation of autologous MSCs effectively repaired corneal wounds, and macrophage depletion completely abrogated MSC-based beneficial effects, confirming that the cooperation between MSCs and macrophages was required for successful vascular regeneration [136]. MSC-injected survivals is dependent on the phenotype and function of tissue-resident macrophages [136,137]. As observed in myocardial infarction and spinal cord in murine models of injury, anti-inflammatory M2 macrophages offer a favorable environment for the engraftment of MSCs [137]. Moreover, the polarization of M1 macrophages to M2 phenotype is critical for the long-term survival of MSCs in healing tissues, suggesting that a reciprocally positive feed-back loop exists between M2 macrophages and MSCs [137]. In a similar fashion, Tregs enhance the survival and engraftment of MSCs in ischemic tissues, and Tregs may even improve the angiogenic properties of MSCs by improving VEGF production [138]. It is important to consider the special characteristics of chronic wound environments, such as low oxygen tension, and how they may influence cell functions. It has been observed that a hypoxic environment diminished macrophage plasticity in response to MSCs [139]. Moreover, in vitro studies showed that macrophages cultured in normoxic conditions with MSCs produced high levels of IL-10, however, while in hypoxic conditions (1% O_2_), the release of the inflammatory cytokine was strongly reduced [139]. In vitro assays showed that MSC from diabetic patients’ adipose tissue demonstrated reduced proliferative capacity and decreased VEGF paracrine release, with lower expression of the stemness gene SOX2 [140]. In keeping, the MSC from Stromal Vascular Faction (SVF) of diabetic patients did not rescue limb ischemiam and this reduced its effect and has been corelated to a significant depletion of CD271+ cells compared to non-diabetic patients [140]. Accordingly, Cianfarani et al. also showed that MSCs from diabetic mice released lower amounts of hepatocyte growth factor and insulin-like growth factor-1 and that the supernatant of diabetic ASCs manifested in a reduced capability to promote keratinocyte and fibroblast proliferation and migration, probably due to a reduce ability for macrophage polarization in M2 [141]. Moreover, the density of adipose-derived cells (ASC) was lower in the adipose tissue of diabetic rats compared with non-diabetic rats and did not promote wound healing in diabetic rats, suggesting that caution is necessary regarding the clinical use of diabetic adipose tissue for the treatment of diabetic wounds [111]. ASC from diabetic patients also exhibited a reduction in VEGF secretion and an impaired angiogenic capacity [112].

Overall, these data suggest that the therapeutic cell therapy potential from the adipose tissue of diabetic patients (SVF, MSC, Adipose derived stem cells ADSC) is dampened when compared with cells isolated from nondiabetic patients because diabetes alters MSCs’ intrinsic properties and impairs their function [142].

In addition, the bone marrow of diabetic patients showed a deep remodeling, consisting of a strong reduction in micro-vessels and sensory neurons, as well as fat accumulation, which creates an unfavorable microenvironment for resident stem cells, which in turn compromises the regenerative efficacy of bone marrow cells which could become harmful vectors of inflammation and anti-angiogenic molecules in diabetic patients [143,144]. This is an important issue that emerging autologous therapies should keep in consideration regarding diabetic non-healing wounds.

### 7.4. Extracellular Vesicles (EVs) and Exosome (Exo)

EV is a generic term for membrane-contained particles naturally released by cells, not containing a nucleus. EVs are traditionally divided into subtypes based on the vesicle sizes: exosomes (50–150 nm diameter), microvesicles (100–1000 nm diameter), and apoptotic bodies (50–4000 nm diameter). Exosomes are formed after the fusion of endosomes membrane with the plasma membrane, while both microvesicles and apoptotic bodies are generated by direct outward blooming from the cell surface [145]. MSC-derived extracellular vesicles (MSC-EVs) can transfer functional proteins and nucleic acids, including microRNAs (miRNAs) and messenger RNAs (mRNAs) to other cells without cell-to-cell contact. Recent studies have demonstrated that MSC-EVs reduce M1 polarization and/or promote M2 polarization in a variety of settings such as cardiovascular, pulmonary, digestive, renal, and central nervous system diseases [145]. An in vitro study revealed that MSCs derived from adipose tissue through exosome release induce M2 polarization [146]. He et al. recently showed that the early depletion of macrophages also delayed wound repair after MSC injection, confirming that MSC-mediated wound healing requires macrophages [147]. In the same paper, the authors demonstrated that MSCs from bone marrow infused systemically could translocate to reach the wound site, promote M2 polarization, and enhance wound healing [147]. The authors also observed that exosomes derived from MSCs induced macrophage polarization while the depletion of the exosomes of MSCs reduced the M2 phenotype [147]. Infusing MSCs without exosomes produced a smaller number of M2 in the wound site and delayed repair [147]. The paper also showed that miR-223, derived from the exosomes of MSCs, regulated macrophage polarization by targeting transcription factor p-knox 1 [147]. These important findings provided evidence for the first time that MSC provokes M2 polarization and could accelerate wound healing by releasing exosome-derived microRNA. Li et al. confirmed that macrophage-derived exosomes exercised anti-inflammatory effects through the inhibition of the secretion of inflammatory enzymes and cytokines and provided the healing of diabetic wound by significantly quickening angiogenesis and improving repair [148]. Another study confirmed that M2-derived exosomes (M2-Exo) induce a complete switch of M1 to M2 [52]. The subcutaneous injections of M2-Exo into the wound edge decreased the local populations of M1 and increased the M2 population and accelerated wound healing by improving angiogenesis, re-epithelialization, and collagen deposition. Accordingly, in a diabetic rat model, it has been observed that exosomes which are overexpressing transcription factor Nrf2 hasten wound healing by inducing vascularization [149]. Exosomes derived from macrophages may represent a novel therapeutic strategy in the treatment of diabetic wound damage.

### 7.5. Dermal Substitutes

Fully acellular dermal substitutes are used in DFU treatment because of the high safety profile and beneficial outcomes as reported in literature. Ideal scaffolds and tissue substitutes including skin matrices should be non-immunogenic, regenerative, protective, durable, and biocompatible. On the basis of the innovative macrophage-centered approach, they also should have a good capacity to induce M2 polarizations [150]. Their therapeutic outcome originates from and is dependent on their source, method of preparation, and further modification. The decellularization method and tissue source can deeply affect the wound microenvironment when the substitute is implanted. Cross-linking or the possible addition of other substances can affect the wound environment and the clinical outcome as well [151]. The chemotactic attractiveness of human fibroblasts to collagens I, II, and III has been studied for many years and is well recognized. Monocytes’ adhesion to collagen types I and III showed a noticeable effect on the secretion of different mediators, including growth factors, cytokines, and enzymes, which in turn play a key role in normative wound healing [152]. Predictably, the diverse surface morphologies and integrated active components can induce an effect on the macrophage’s phenotype. Consequently, it is extremely important to study the immunomodulatory effects of dermal substitute, especially when implanted on chronic and/or diabetic wound. It has been observed that particular geometrical parameters could direct human macrophage polarization [153]. Fibrous collagen scaffolds with box-shaped pores and precise inter-fiber spacing from 100 μm down to only 40 μm facilitate primary human macrophage elongation accompanied by differentiation towards the M2 type [153]. Table 2 show commercially available dermal substitutes evaluated for their immunomodulatory and M2 polarization ability.

Yin et al. found that the pore size of a scaffold influences the phenotypes of resident macrophages, showing that a relatively larger size (~360 μm) leads to enhanced formation of blood vessels, with higher levels of VEGF+ cells and a lower level of M1 macrophages [154]. In addition to pore size, collagen-functionalizing additives may also have an effect on macrophage activation, such as Chondroitin sulfate (CS), which has been reported to play vital roles in the immune response. CS at an increasing dose range of 100–1000 μg/mL was found to significantly increase the phagocytic activity and ROS production as well as the secretion levels of NO, TNF-α, IL-6, and IL-10 by monocyte/macrophage lineage (RAW264.7) [155]. Witherel et al. studied the responses of monocyte-derived macrophages isolated from blood to four different commercially available biomaterials in vitro: OASIS^®^ Wound Matrix, which is an extracellular matrix from porcine small intestinal mucosa; INTEGRA^®^ Bilayer Matrix, a dermal bilayer of cross-linked bovine tendon type I collagen and chondroitin-6-sulfate plus a layer of polysiloxane; AlloMend^®^ Acellular Dermal Matrix, a decellularized matrix composed mainly of collagen and elastin; and PriMatrix^®^ Dermal Repair Scaffold, decellularized fetal bovine dermis rich in type I and II collagen [151]. The OASIS^®^ and INTEGRA™ matrices downregulated the expression of M2a anti-inflammatory markers CCL22 and TIMP3, suggesting a probable inhibition of extracellular matrix secretion and fibrosis, which are crucial events for wound closure. OASIS^®^ was also the biomaterial responsible for the greatest increase in M1 genes expression. The authors suggest that INTEGRA^®^ inflammatory response could be related to glutaraldehyde cross-link and suggest that both OASIS^®^ and INTEGRA™ seem to be a poor option for chronic wounds [151]. PriMatrix^®^ as well showed a downregulation of the anti-inflammatory genes CCL22 and TIMP3 and an overexpression of both the pro- inflammatory cytokine TNF-α and CD163, associated with M2c. AlloMend^®^ only induced an effect of the upregulation of CD163, and it was considered the biomaterial with the lowest influence on macrophage response. Agrawal et al. compare DermaMatrix^®^, AlloDerm^®^, Integra^®^, and DermACELL^®^ M1/M2 polarization in an animal model [156]. Macrophage surface markers CD68 (all macrophages), CCR7 (M1 phenotype), and CD206 (M2 phenotype) were used to characterize an M1–M2 profile by an immuno-histological assay. All dermal substitutes showed a bell-shaped curve for the distribution of CD68+ macrophages, except Integra^®^, which showed an increasing trend of macrophages with time [156]. Moreover, DermACELL^®^ had the highest entry of macrophages, while Integra^®^ had the smallest [156]. AlloDerm^®^ showed that the macrophages were mostly M1 at 7, 14, 21, and 42 days post implantation, while Integra^®^ showed a mixed M1/M2 population of macrophages at all time-points: The trend for the M1:M2 ratio was skewed towards M2 on day 7, towards M1 on days 14 to 21, and again towards M2 on day 42 for Integra^®^ [156]. A recent study showed that the implant of a porcine urinary bladder matrix (UBM) is associated with the modulation of wound inflammation in diabetic patients, measured as mRNA associated with M1 and M2 macrophages [157]. Recently, Montanaro et al. show investigate how the dermal substitute Nevelia^®^, which is a dermal substitute consisting of a three-dimensional porous matrix of type 1; purified, stabilized, bovine-origin collagen; and a layer of reinforced silicone may influence the inflammatory infiltrate and macrophages polarization [158]. The study randomly enrolled 15 diabetic patients with chronic foot ulcers, 5 treated only by standard of care as control group, and 10 treated with Nevelia^®^. Biopsy was performed at baseline and after 30 days and histological, immunohistochemical, and immunofluorescence analysis was performed to evaluate the number of M1 and M2 macrophages. Dermal substitute group showed a general macrophage activation and a greater and significative polarization toward M2 subpopulation at 30 days, compared with control. The increase in M2 phenotypes population was also confirmed by confocal microscopy. Moreover, after 6 months, 6 patients (60%) of the Nevelia^®^ completely healed, while only 1 patient (20%) healed in the control group, suggesting that this dermal substitute induce tissue reparative processes through macrophage activation and M2 reparative polarization in diabetic lesions [158]. The positive clinical outcome of this dermal substitute was previously observed by the sane authors in 41 patients with chronic diabetic wound [159]. In addition, Nevelia^®^ dermal substitute was observed to polarize in M2 and also in an in vitro model [160].

**Table 2 jcm-11-00889-t002:** Dermal Substitutes tested for immunomodulatory and macrophage polarization ability.

	Primary Material Composition	Source and Other Components	Refs.
Nevelia	Porous resorbable double layer matrix 2 mm thickness made of stabilized native collagen type I and a silicone sheet 200 mm thickness mechanically reinforced with a polyester fabric. The extraction procedure and the freeze-drying process allow the structuring of the collagen into a matrix with optimal hydrophilicity, pore structure and pore size (20–125 μm)	Bovine, Native collagen Type I. No glycosaminoglycan (GAG) added to improve cell attachment and proliferation. Glutaraldehyde Cross-linking	[157,158,159]
Integra	Bilayer system for skin replacement made of a porous matrix of fibers of cross-linked bovine tendon collagen and glycosaminoglycan (chondroitin-6-sulfate) that is manufactured with a controlled porosity and defined degradation rate. The Integra pore size of 20 to 125 μm allows influx of cells.	Bovine Tendon Type I Collage Shark cartilage -derived chondroitin-6-sulphate (GAG). Glutaraldehyde Cross-linking	[149,150,155]
PriMatrix	Acellular dermal tissue matrix. comprising of both type I and type III collagen derived from fetal bovine dermis. This matrix is processed in a way to maintains the extracellular matrix in its native and undamaged state while removing all lipids, fats, cells, carbohydrates and non-collagenous proteins.	Fetal Bovine collagen type I and type III collagen. No cross-link	[150]
Oasis Wound Matrix	Lyophilized, decellularized porcine small intestine submucosa (SIS). Matrix is derived from a single layer of porcine small intestinal submucosa (SIS) technology. The technology provides an intact three-dimensional extracellular matrix which allows for host cell migration. The SIS is freeze-dried and sterilized with ethylne oxide gas in preparation for clinical use	Porcine small intestine submucosa (SIS). No cross-link	[150]
Allomend	Decellularized donated human dermal tissue, with significant removal of cellular debris (including DNA and RNA), proteins and antigens. The process does not require the use of detergents or enzymes, thereby mitigating the possibility of harmful residuals in the tissue. The decellularization process also inactivates microorganisms through cellular disruption. USA only, not available in Europe	Human dermal tissue No cross link	[149,150,155]
DermaMatrix	Cadaveric human allograft treated with a disinfectant solution that combines detergents with acidic and antiseptic reagents. USA only, not available in Europe	Human dermal tissue No cross link	[149,150,155]
Dermacell	Decellularized regenerative human tissue matrix allograft processed using proprietary technology that removes at least 97% of donor DNA without compromising the desired biomechanical structure or biochemical properties. USA only, not available in Europe	Human dermal tissue No cross link	[149,150,154]

## 8. Conclusions

Both sustained increases in the number of wound macrophages together their phenotype dysregulation towards the inflammatory types, caused by intrinsic alterations in bone marrow and by a pro-inflammatory wound microenvironment, cause impaired wound healing in diabetes. Our understanding of the macrophages populations during impaired healing is still partial. Diabetic wounds with potentially devastating consequences on suffering patients remain a strong medical need. The understanding of the systemic and local immune responses is fundamental to develop innovative therapies. Moreover, it could be useful to verify how current therapy influence macrophage polarization to identify better treatment for chronic wounds in diabetic individuals. Each patient’s immune system represents a dynamic history of infections, sex, age, diet, genetic characteristics, and environmental factors. Innovative therapy should be designed to manipulate the immune system to switch towards anti-inflammatory and regenerative phenotype that promotes the desired repair outcome.

## Figures and Tables

**Figure 1 jcm-11-00889-f001:**
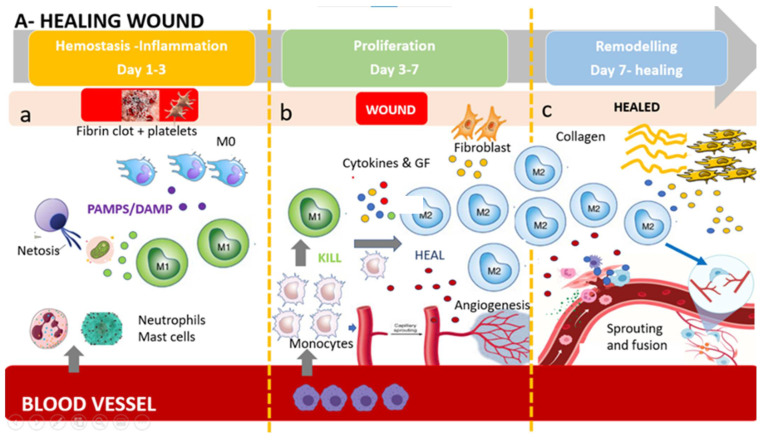
The healing wound: (**a**) Platelets form a fibrin clot, and chemo-attractants are released to recruit inflammatory cells (neutrophils and mast cells), releasing pro-inflammatory cytokines. NETosis (Neutrophil Extracellular Trap) helps to capture and destroy pathogens. Tissue-resident macrophages react to pathogen- and damage-associated molecular patterns (PAMPs and DAMPs). First wave of monocytes differentiates in M1(phagocytoses step). (**b**) After the resolution of inflammation, the proliferative phase starts. Angiogenesis develops via vessel sprouting. Infiltrating monocytes differentiate into M1 and M2 macrophage subsets. M1 macrophages maintain a strong inflammatory profile releasing inflammatory cytokines and ROS and eating dead bacteria and neutrophils. After M1 polarize in M2 pro-regenerative phenotype which release anti-inflammatory cytokines, growth factors, and proteases which replace the provisional ECM with collagens, induce fibroblasts proliferation, and induce new vessel formation. This process results in granular tissue and keratinocyte coverage. (**c**) Remodeling is supported by macrophages, fibroblasts, and myofibroblasts re-organizing the provisional ECM into a definitive healed tissue, principally through matrix metalloproteinases (MMPs) and their inhibitors (TIMPs), resulting in tissue with strong tensile strength and functionality. Angiogenesis is almost complete, and macrophages produce molecule bypass (fusion) between newly formed vessels, creating a functional network.

**Figure 2 jcm-11-00889-f002:**
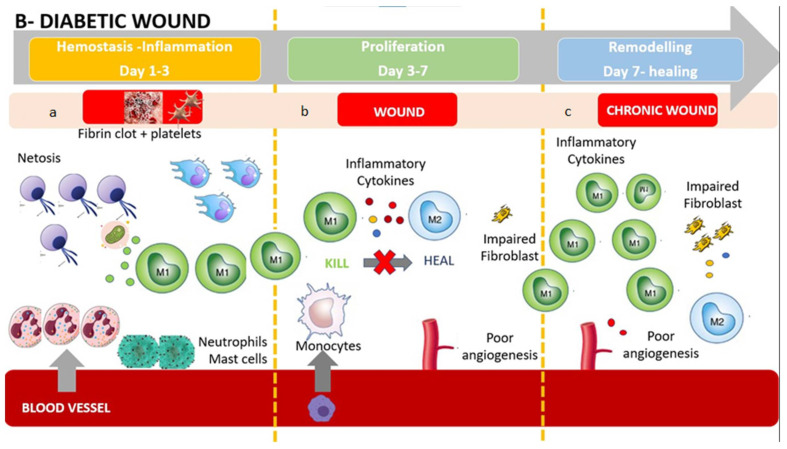
The diabetic wound: (**a**) Impaired wounds showed an upregulated influx of neutrophils and mast cells, leading to an intense inflammatory response, causing collateral damage, and extending the inflammatory phase to subsequent phases. The persistent higher release of inflammatory cytokines produces M1 activation with further release of inflammatory substances. (**b**) Monocyte recruitments are poor due to arterial occlusion and impaired microcirculation. Poor angiogenesis and glycated proteins result in an impaired fibroblast activity. The hypoxic environment brings oxidative stress, driving inflammatory M1 macrophage polarization and impairment of fibroblasts, resulting in poor ECM reorganization and a persistent inflammatory environment. The polarization in M2 is absent or extremely poor, causing a further accumulation of M1. (**c**) Impaired wound-resident cells remain ineffective and in an inflammatory condition. Collagen reorganization resolves poorly, resulting in weak, non-functional skin that can re-injure and potentially ulcerate, perpetually inflamed. Macrophages are still activated in the inflammatory phenotype M1. The wound does not heal.

**Figure 3 jcm-11-00889-f003:**
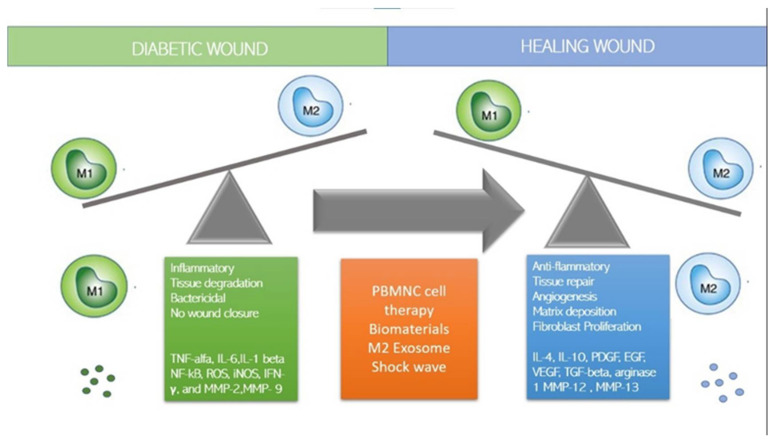
How to switch to M2: strategies.

**Table 1 jcm-11-00889-t001:** Immune-Cell-based Cell Therapy—Clinical trials on diabetic patients.

	Description	Result	Ref.
Zuloff-Shani et al. 2004	Treatment of chronic ulcers with blood-derived macrophages activated by hypo-osmotic shock in over 1000 patients	Reduction of the healing time, reduction of risk of complications and morbidity. Improvement of the quality of life for long-suffering patients	[107]
Danon et al.	Decubital ulcers of 72 patients (average age 82), were treated by local injection of macrophages prepared from a blood unit in a closed sterile system. The remaining 127 patients (average age 79) were treated conventionally and served as controls. No exclusion criteria were applied.	In the macrophage-treated group, 27% healed, while only 6% healed in the control group (*p* < 0.001). Moreover, the macrophage-treated group showed a faster healing (*p* < 0.02)	[113]
Zuloff-Shani et al. 2010	100 consecutive elderly patients with a total of 216 stage III or IV pressure ulcers, 66 patients were assigned to the autologous macrophages group, 38 patients were assigned to the standard care treatments (38 patients.)	Percentage of completely closed wounds (wound level and patient level) were significantly better (*p* < 0.001/*p* < 0.001, respectively) in all patients in favor of AMS, as well as in the subset of diabetic patients (*p* < 0.001/*p* < 0.001).	[114]
Moriya, J et al.	Retrospective study on 42 patients with severe intermittent claudication, ischemic rest pain, or non-healing ischemic ulcers caused by peripheral arterial disease, including thromboangiitis obliterans, and who had not responded to conventional therapy that included nonsurgical and surgical revascularization (no option).	Improvement of ischemic symptoms was observed in 60% to 70% of the patients. The annual rate of major amputation was decreased significantly by treatment. The survival rate of younger responders was better than that of non-responders.	[116]
Huang, P.P et al.	150 patients with peripheral arterial disease were randomised to mobilized PBMNC 76 cases or BMMNC 74 cases implanted, follow up for 12 weeks. Primary outcomes were safety and efficacy of treatment, based on ankle-brachial index (ABI) and rest pain	Significant improvement of the ABI, skin temperature and rest pain was observed in both groups after transplantation and was better I PBMNC group. However, there was no significant difference between two groups for pain-free walking distance, transcutaneous oxygen pressure, ulcers, and rate of lower limb amputation	[117]
Liotta, F et al.	Autologous Non-Mobilized Enriched Circulating Endothelial Progenitors obtained from non-mobilized peripheral blood by immunomagnetic selection of CD14+ and CD34+ cells) or BM-MNC were injected into the gastrocnemius of the affected limb in 23 and 17 patients with no option critical limb ischemia.	After 2 yrs follow-up, both groups showed significant and progressive improvement in muscle perfusion (primary endpoint), rest pain, consumption of analgesics, pain-free walking distance, wound healing, quality of life, ankle-brachial index, toe-brachial index, and transcutaneous PO2	[118]
Dubsky, M et al.	28 patients with diabetic foot disease (17 treated by bone marrow cells and 11 by peripheral blood mononuclear cell) were included into an active group and 22 patients into a control group without cell treatment.	The transcutaneous oxygen pressure increased significantly (*p* < 0.05) compared with baseline in both active groups after 6 months, with no significant differences between bone marrow cells and peripheral blood cell groups, while no change in the control group was observed. The rate of major amputation by 6 months was significantly lower in the active cell therapy group compared with that in the control group (11.1% vs. 50%, *p* = 0.0032), with no difference between bone marrow cells and peripheral blood cells.	[119]
Persiani, F. et al.	50 diabetic patients affected by CLI underwent PBMNCs implant (32 patients underwent PBMNCs therapy associated with endovascular revascularization, 18 patients, non-option CLI)	The follow-up period was 10 months. In the PBMNC group + revascularization TcPO, pain VAS Scale improved. In PBMNCs therapy group, the mean TcPO2 improved from 16.2 ± 7.2 mmHg to 23.5 ± 8.4 mmHg (*p* < 0.001), and VAS score means decreased from 9 ± 1.1 to 4.1 ± 3.3 (*p* < 0.001). Major amputation was observed in 3 cases (9.4%), both in adjuvant therapy group and in PBMNCs therapy. (16.7%) (P ¼ 0.6) as the therapeutic choice (PBMNCs therapy group).	[120]
De Angelis, B et al.	Prospective, not randomized study based on a treated group who did not respond to conventional therapy (n = 43) when implanted with A-PBMNC cells versus a historically matched control group. Patients of both groups were suffering from CLI Fontaine scale IV with chronic ulcers	The A-PBMNC-treated group showed a statistically significant improvement of limb rescue of 95.3% versus 52.2% of the control group (*p* < 0.001) at 2 years. The A-PBMNC group also showed reduction in pain at rest, increased maximum walking distance, and healing of the wound and an overall improvement in the quality of life. Post-treatment radiological studies showed an improvement of vascularization with the formation of new collateral and by histological findings.	[121]
Dubsky et al.	31 patients with DFU and CLI treated by autologous stem cells and 30 patients treated by PTA were included in the study; 23 patients with the same inclusion criteria who could not undergo PTA or cell therapy formed the control group.	Amputation-free survival after 6 and 12 months was significantly greater in the cell therapy and PTA groups compared with controls (*p* < 0.001 and *p* < 0.0029, respectively) without significant differences between the active treatment groups. Increase in TcPO2 did not differ between cell therapy and PTA groups until 12 months but TcPO2 in the control group did not change over the follow-up period. More healed ulcers were observed up to 12 months in the cell therapy group compared with the PTA and control groups (84% vs. 57.7% vs. 44.4%; *p* < 0.042).	[127]
Scatena et al.	The study included 76 NO-CLI patients with DFUs. All patients were treated with the same standard care (control group), but 38 patients were also treated with autologous PBMNC implants.	Only 4 out 38 amputations (10.5%) were observed in the PBMNC group, while 15 out of 38 amputations (39.5%) were recorded in the control group (*p* = 0.0037). The Kaplan–Meier curves and the log-rank test results showed a significantly lower amputation rate in the PBMNCs group vs. the control group (*p* = 0.000). At two years follow-up, nearly 80% of the PBMNCs group was still alive vs. only 20% of the control group (*p* = 0.000). In the PBMNC group, 33 patients healed (86.6%) while only one patient healed in the control group (*p* = 0.000).	[129]
Di Vieste et al.	Case report of a 59-year-old patient with type 2 diabetes mellitus who had a gangrene of the right toe. After an ineffective angioplasty, it was decided to use a PBMNC therapy.	The patient underwent to amputation of the first necrotic toe and three PBMNC treatment sessions with complete surgical wound healing and limb rescue	[130]

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
