# Peer review of "The Immune-Centric Revolution in the Diabetic Foot: Monocytes and Lymphocytes Role in Wound Healing and Tissue Regeneration—A Narrative Review"

_jcm, 2022, doi:10.3390/jcm11030889_

Round 1
Reviewer 1 Report
The current review summarizes the regulatory functions of innate and adaptive immune cells at different stages of normal skin wound healing and describes how deviation from this may lead to impaired healing of diabetic wounds both in humans and animal models. This is an important topic, and the current review provides a thorough and updated information with 178 references. However, the review may improve if the following comments/suggestions are taken care of.
Comments
Macrophage classification based on activation cues and surface markers can be combined as they appear to be repetitive.
Similarly, there are some repetitions throughout the manuscript including the T-reg section which could be concise.
There are descriptions of other tissue damage in addition to diabetic foot ulcer in few places which may divert the readers from the main topic.
“In a healing wound ….. , phagocytosis of microbes and necrotic damaged tissue “ (L356-359). This is a complicated sentence and hard to follow.
Seems like the last paragraph in section “6. Monocytes/macrophages and limphocytes in diabetic wound healing” is unrelated to the current topic. Also, the authors talk about monocyte/macrophages followed by neutrophils and Lymphocytes and then monocytes/macrophages again in this section. It would be good put all the monocyte/macrophage information together in the beginning.
There is too much information in the cell based and other therapy sections. It would be helpful for the readers if all the information is also summarized in a table.
Author Response
Macrophage classification based on activation cues and surface markers can be combined as they appear to be repetitive.
- Thank you for the suggestion, we combined activation cues and surface markers in a single chapter
Similarly, there are some repetitions throughout the manuscript including the T-reg section which could be concise.
- T-reg section it has been reduced and repetition eliminated
There are descriptions of other tissue damage in addition to diabetic foot ulcer in few places which may divert the readers from the main topic.
- We eliminated descriptions of other tissue damage (for example, Treg in Duchenne syndrome it has been eliminated)
“In a healing wound ….. , phagocytosis of microbes and necrotic damaged tissue “ (L356-359). This is a complicated sentence and hard to follow.
- The sentence was correct in “ In a healing wound (Fig.1A), just after neutrophils recruitment, a first wave of monocytes invades the tissue, differentiate into inflammatory M1 macrophages, releasing cytotoxic and proinflammatory molecule such as IL-1β, TNF-α, IL-6 and ROS, with the aim to digest damaged cell, microbes and necrotic damaged tissue.”
Seems like the last paragraph in section “6. Monocytes/macrophages and limphocytes in diabetic wound healing” is unrelated to the current topic.
- Thank you for the suggestion, this was a mistake. This part it has been moved in section 7.2. Immune Cells based -cell Therapy
Also, the authors talk about monocyte/macrophages followed by neutrophils and Lymphocytes and then monocytes/macrophages again in this section. It would be good put all the monocyte/macrophage information together in the beginning.
- Thank you for the suggestion, we correct the section and we also split it in four part:Neutrophils in diabetic wound, Monocytes-macrophages in diabetic wound, Lymphocytes in diabetic wound, Keratinocytes and fibroblasts in diabetic wound
There is too much information in the cell based and other therapy sections. It would be helpful for the readers if all the information is also summarized in a table.
- Thank you for the suggestion. We added Table I for autologous cell therapy based in immune cells and Table II for dermal substitute.
Reviewer 2 Report
This present review discusses in detail about immunological perspective on wound healing in both normal and diabetic conditions. This review speaks about a better understanding of the role of M1 and M2 macrophage in both normal and diabetic condition wound healing. Most importantly this review points out poor angiogenesis and M2 to M1 switch in diabetic wounds. I have few suggestions to improve the manuscript for publication.
Minor comments:
Spelling mistakes detected in a number of places for example lymphocytes mistakenly written as limphocytes.
Pixel quality of image provided needs to be improved, in figure 1, it's very hard to read capillary sprouting.
Acronym DFU was used but its definition was not mentioned in the text. Likewise, acronym NO-CLI was mentioned in line number 575, however partial definition was written in line number 629.
Last section on Dermal substitute, if they can provide small table on Matrix used and its clinical outcomes so that readers can easily understand.
Author Response
Minor comments:
Spelling mistakes detected in a number of places for example lymphocytes mistakenly written as limphocytes.
- Corrected, thank you.
Pixel quality of image provided needs to be improved, in figure 1, it's very hard to read capillary sprouting.
- We enlarge the image, we also cancel the “capillary sprouting” label. .
Acronym DFU was used but its definition was not mentioned in the text. Likewise, acronym NO-CLI was mentioned in line number 575, however partial definition was written in line number 629.
- DFU an NO-CLI have been defined ,thank you for the suggestion
Last section on Dermal substitute, if they can provide small table on Matrix used and its clinical outcomes so that readers can easily understand.
- We added Table 1 on dermal substitute tested for immunomodulatory ability.
Round 2
Reviewer 1 Report
The authors have revised the manuscript according to the comments. The manuscript can be accepted for publication now.